# Trends in Antituberculosis Drug Resistance and Associated Factors: A 31-Year Observational Study at a Tertiary Hospital in Barcelona

**DOI:** 10.3390/antibiotics14090875

**Published:** 2025-08-30

**Authors:** Mateu Espasa, Belén Pagán, Mariana Fernández-Pittol, Ángels Orcau, Griselda Tudó, Felipe García, Jose-Antonio Martínez, Néstor Soler, Laura Horvath-Ruiz, Lorena San-Nicolás, Diego Martínez, Climent Casals-Pascual, Jordi Vila, Juan-Pau Millet, Joan A. Caylà, Julian Gonzalez-Martin

**Affiliations:** 1Servei de Microbiologia, CDB, Hospital Clínic de Barcelona, c/Villarroel 170, 08036 Barcelona, Spain; mespasa@clinic.cat (M.E.); mjfernandez@clinic.cat (M.F.-P.); lhorvath@clinic.cat (L.H.-R.); lsannic@clinic.cat (L.S.-N.); dmartine@clinic.cat (D.M.); ccasals@clinic.cat (C.C.-P.); jvila@clinic.cat (J.V.); 2ISGLOBAL, Barcelona Institute for Global Health, c/Rosselló 132, 08036 Barcelona, Spain; griselda.tudo@ub.edu; 3Departament de Fonaments Clínics, Facultat de Medicina i Ciències de la Salut, Universitat de Barcelona, c/Casanova 143, 08036 Barcelona, Spain; brodri27@xtec.cat; 4Agència de Salut Pública de Barcelona, Pl. Lesseps, 08023 Barcelona, Spain; aorcau@gmail.com (Á.O.); jmillet@aspb.cat (J.-P.M.); 5Servei de Malalties Infeccioses, Hospital Clínic de Barcelona, 08036 Barcelona, Spain; fgarcia@clinic.cat (F.G.); mararturketer@hotmail.com (J.-A.M.); 6CIBER of Infectious Diseases (CIBERINFEC), Instituto de Salud Carlos III, 28029 Madrid, Spain; 7Institut d’Investigacions Biomèdiques August Pi i Sunyer (IDIBAPS), c/Rosselló 132, 08036 Barcelona, Spain; nsoler@clinic.cat; 8Departament de Medicina, Facultat de Medicina i Ciències de la Salut, Universitat de Barcelona, c/Casanova 143, 08036 Barcelona, Spain; 9Department of Pneumology, Hospital Clínic de Barcelona, 08036 Barcelona, Spain; 10CIBER de Epidemiología y Salud Pública (CIBERESP), 28029 Madrid, Spain; 11Tuberculosis Research Unit Foundation of Barcelona (fuiTB), 08023 Barcelona, Spain; joan.cayla@uitb.cat

**Keywords:** tuberculosis, drug-resistance, surveillance, immigration, public-health

## Abstract

**Objective**: To analyze trends in resistance to antituberculous drugs over a 31-year period (1991–2022) at a hospital in Barcelona and to identify associated epidemiological determinants. **Methods**: This study included culture-confirmed tuberculosis cases diagnosed between 1991 and 2022. Drug susceptibility testing was conducted with clinical data from hospital records and epidemiological data from the Barcelona Public Health Agency. The primary outcome was resistance to first-line drugs. A subset of isolates was tested for second-line drugs. Trends were compared between the periods 1991–2000 and 2001–2022, aligning with increased immigration. Factors associated with resistance were examined using multivariate regression analysis. **Results**: Among the 2448 patients included, tuberculosis cases peaked in the 1990s and subsequently declined, while drug resistance increased. Overall, 12.2% of isolates showed resistance to at least one drug: 8.5% were monoresistant, 2.3% multiresistant, and 1.4% polyresistant. The 2001–2022 period had a higher resistance rate (OR 1.63; 95%CI 1.28–2.09) but lower multiresistance (OR 0.40; 95%CI 0.23–0.69). Resistance among new cases doubled from 6.4% to 12.8%, while rates among previously treated cases remained stable. The predictors of resistance were foreign-born (OR 1.52; 95%CI 1.21–1.91) and previous tuberculosis treatment (OR 2.88; 95%CI 2.17–3.81). A total of 90% of isolates remained susceptible to fluoroquinolones and aminoglycosides. **Conclusions**: Although tuberculosis incidence has declined over the past three decades, antibiotic resistance has increased, driven by foreign-born and retreatment cases. Ongoing drug susceptibility testing, access to second-line therapies, and targeted public health interventions for high-risk populations are essential to maintain control in low-incidence settings.

## 1. Introduction

Tuberculosis (TB) remains a major global health challenge, despite ongoing international efforts to reduce its burden. Tuberculosis is the 10th leading cause of death worldwide, and the WHO End TB Strategy set the goal of reducing tuberculosis incidence by 50% and tuberculosis deaths by 75% by 2025 compared to 2015 levels [1]. However, the COVID-19 pandemic caused an 18% drop in tuberculosis case notifications, a 7% increase in mortality in 2020, and a 15% decrease in access to treatment for multidrug-resistant tuberculosis [2]. In Spain, the tuberculosis incidence rate dropped by 16.6% in 2020, and between 2018 and 2022, it decreased by 6.5%. Despite this, achieving the 50% reduction goal by 2025 in Spain remains unlikely, as it happens in other Western European countries, including Portugal, Italy, France, and Belgium [3].

In 2022, Spain reported a tuberculosis incidence rate of 8.07 cases per 100,000 population. In Catalonia, the rate was higher at 12.6 per 100,000, with a total of 1017 cases. Of these, 72% were pulmonary, and 58% occurred in foreign-born individuals. Drug-resistant tuberculosis represented 8.9% of all cases, including 1.4% classified as multidrug-resistant tuberculosis (MDR-tuberculosis) [3].

In 2023, an estimated 400,000 people worldwide were affected by rifampicin-resistant or MDR-tuberculosis. To address this, a United Nations meeting set a global target to achieve 90% treatment coverage for both drug-susceptible and drug-resistant TB cases by 2027 [1]. An important aspect to reach this goal is the surveillance of drug-resistant *M. tuberculosis* strains that is based on phenotypic drug susceptibility testing as the reference standard. This will allow us to know the epidemiological situation of TB resistance and obtain the best treatment for patients.

Tuberculosis treatment consists of a six-month course of four drugs, each targeting different aspects of bacterial metabolism, to prevent the development of resistance. Resistance primarily emerges through chromosomal mutations in drug targets, often driven by inadequate treatment, poor adherence, or the transmission of resistant strains [4]. In 2023, WHO data indicated that 3.2% of new cases and 16% of previously treated cases had MDR-tuberculosis, with 6.2% of those being extensively drug-resistant [XDR-tuberculosis) [1]. The treatment for resistant cases requires second-line drugs, which are more toxic, less effective, and require longer treatment durations [4,5]. Recently, new drugs have been incorporated, and the WHO and the Spanish societies of pulmonary (SEPAR) and infectious diseases and clinical microbiology (SEIMC) defined recommended treatment schedules for resistant cases [5,6].

This study aims to analyze the evolution of resistance to first- and second-line anti-tuberculosis drugs at a tertiary hospital in Barcelona over 31 years (1991–2022), focusing on resistance patterns and sociodemographic factors influencing its evolution. The results would help to know the TB control program outcomes and identify TB resistance risk factors and improve its management.

## 2. Results

### 2.1. Global TB Cases Distribution by Period

A total of 2534 cases were recorded. A total of 86 cases were excluded for incomplete data, leaving 2448 for the analysis: 1301 cases from 1991 to 2000 and 1147 cases from 2001 to 2022.

As shown in Figure 1, the total number of tuberculosis cases declined over time (y = −3.83x + 7763.55, R^2^ = 0.83, *p* < 0.05), while resistance rates increased (y = 0.56x − 1109.00, R^2^ = 0.49, *p* < 0.05), and MDR-tuberculosis strains slightly declined (y = −0.09x + 187.32, R^2^ = 0.20, *p* < 0.05).

### 2.2. TB Cases Demographic Data

Significant differences were observed between the two periods (Table 1). In the second period, there was an increase in women, median age, homeless people, foreign-born patients, comorbidity, and cure rate, but a decrease in HIV positivity, deaths, and smear-positive samples.

Between 1991 and 2000, 56.9% of patients completed treatment, compared with 69.9% in 2001–2022 (*p* < 0.001). Mortality decreased from 17.4% to 9.7% (*p* < 0.001), and both treatment abandonment and loss to follow-up were also significantly reduced in the later period. Overall, these results demonstrate a marked improvement in treatment outcomes over time.

One-quarter of patients were foreign-born, although there were differences between both periods, rising to 45% in the second. Tuberculosis cases among foreign-born individuals increased across all geographic regions, especially from Latin America and Asia (Figure 2). In 1991–1995, Latin American and Asian patients represented 4.8% and 1.9% of tuberculosis cases, respectively, rising to 29.9% and 18.2% in 2016–2022. Africa–Maghreb countries also rose (3.1% to 10.3%), and other European countries rose from 0.2% to 7.5%.

### 2.3. 1st Line TB Drug Resistance Data

Overall, 12.2% of patients showed resistance to any drug; isoniazid (7.8%) and streptomycin (3.4%) were the most frequent. Monoresistance was 8.5%, multiresistance 2.3%, and polyresistance 1.4% (Table 2). Comparing both periods, there was a 63.3% increase in overall resistance (8.6% to 16.3%), mainly monoresistance (5.5% to 11.9%) and polyresistance (0.4% to 2.6%), while MDR-tuberculosis decreased (2.8% to 1.7%).

Between both periods, new cases increased from 85.3% to 94.1%, and overall resistance doubled (from 6.4% to 12.8%). Rifampicin resistance was the only one that decreased between periods (Table 2).

The resistance by region and period (Figure 3) showed a monoresistance increase in Africa–Maghreb (10.9% vs. 22.2%) and Asia (13.5% vs. 20.7%). Latin America maintained high monoresistance (13.0%) with small increases in multiresistance and polyresistance. Spain showed a rise in monoresistance (4.5% vs. 7.2%) but decreased multiresistance (2.2% vs. 0.6%).

The distribution of resistance patterns in new and previously treated tuberculosis cases reveals significant differences in monoresistance, multiresistance, and polyresistance to key antimycobacterial agents (Appendix A). Among new cases, resistance to isoniazid was the most frequent (6.3%), being higher among foreign-born (11.4%) compared to Spanish-born cases (4.3%). Streptomycin monoresistance was also notably higher in foreign-born new cases (6.6%) than in Spanish-born cases (1.4%). Previously treated cases had a significantly higher proportion of resistance across all drugs, particularly for isoniazid (44.2% of foreign-born compared to 14.8% of Spanish-born). This was also observed in rifampicin resistance (20.9% foreign-born versus 7.0% Spanish-born). Multiresistance patterns were more common in previously treated cases, with combinations such as H + R (4.7%) and H + R + Z (5.1%) observed primarily in foreign-born. Although rare, resistance to four or five drugs was almost exclusively detected in foreign-born cases, particularly in new cases, where resistance patterns such as H + R + E + Z + S (0.2%) were found. Streptomycin resistance increased mostly in Eastern Europe (0% to 26.9%) and Latin America (1.8% to 10.9%), for both low and high levels of resistance (Appendix A).

Multivariate analysis showed that foreign-born status tripled the OR and previous tuberculosis treatment doubled it. Despite an increase in TB-resistance cases after 2000, with a 1.63-fold higher risk, it was not statistically significant in the multivariate analysis, probably influenced by the foreign-born status and previous TB treatment. (Table 3).

### 2.4. 2nd Line TB Drug Resistance Data

Second-line DST was performed in 205 isolates showing resistance to any first-line drug, and most second-line agents showed high efficacy (Table 4). Ethionamide (16.4%) and cycloserine (9.4%) had the highest overall resistance rates. Ethionamide is notably associated with MDR-tuberculosis strains (41.9%). Rifabutin showed a 100% resistance, all among MDR-tuberculosis strains, due to rifampicin cross-related resistance. In contrast, resistance to other second-line agents—amikacin, capreomycin, kanamycin, and linezolid—remained low.

The second-line drugs were also studied in the subset of first-line susceptible isolates, according to the following distribution: ethionamide in 104 isolates, amikacin and kanamycin in 105, capreomycin in 103, ofloxacin in 102, ethionamide in 104, cycloserine in 99, and rifabutin in 14. The analysis showed 0.96% resistance to ethionamide, 0.98% to ofloxacin, and 4% to cycloserine. No resistant isolates were observed for rifabutin, amikacin, kanamycin, and capreomycin.

## 3. Discussion

This study examines drug resistance in patients with tuberculosis over a 31-year period at a tertiary hospital, comparing demographic, clinical, and resistance-related factors across two distinct time frames: the HIV period and the immigration-driven period. Statistically significant differences were observed between the two periods in terms of sex, age, underlying conditions, and overall drug resistance. The findings underscore that drug resistance remains a critical challenge, particularly among foreign-born individuals and those with a history of previous treatment, despite a general decline in tuberculosis cases. Continued monitoring of resistance patterns, sustained investment in TB control programs, improved treatment adherence, and strong epidemiological surveillance remain essential for effective disease management.

Overall, 11.5% of patients with tuberculosis exhibited resistance to at least one first-line drug, with resistance rates ranging from 9.8% in new cases to 20.5% in previously treated cases. When analyzed by period, resistance ranged from 8.3% (1991–2000) to 13.6% (2001–2022). These results align with other studies conducted in Spain (rates 3.9–11.7%) [7,8,9], describing higher percentages in studies from 2000 onwards [9]. Our data suggest that much of the increase is linked to foreign-born patients, with Latin America and Pakistan most prevalent, followed by Eastern Europe.

Isoniazid was the drug with the highest resistance rate (6.2% overall) but rose to 12.4% in foreign-born new cases and 44.2% in previously treated foreign-born patients. This likely reflects higher rates of resistance in their countries of origin. Similar patterns were seen for other drugs. Our isoniazid resistance matches some multicenter local studies [10,11]. Because isoniazid resistance can reduce standard treatment success and foster MDR-tuberculosis, controlling it remains crucial [12].

Streptomycin resistance mainly explains the global resistance rise among foreign-born new cases in the second period. Aside from methodological changes, it also reflects higher prevalence among immigrants (Appendix A). This prevalence can be related to the use of this antibiotic in the past in the country of origin. Even though streptomycin is not part of standard regimens, it can be a useful alternative to aminoglycoside in certain cases, as can amikacin.

Among resistance categories, monoresistance (one single first-line drug) was most frequent, followed by MDR-tuberculosis (H and R drug resistance) and polyresistance (two or more drugs resistance not including H and R). From 1991 to 2000 to 2001–2022, monoresistance doubled and polyresistance increased sevenfold, whereas MDR-tuberculosis halved. Although some regions, as Eastern Europe [13], face alarming MDR-tuberculosis rates, our rate (2.3%) declined in the second period, consistent with other Spanish studies [7,11,14,15] but differing from, for instance, one study in Asturias [10] reporting much lower MDR-tuberculosis (0.37%) despite higher tuberculosis incidence.

The decline in MDR-tuberculosis may be attributed to sustained tuberculosis control efforts over recent decades, including the Barcelona Tuberculosis Prevention and Control Program [16,17]. This initiative has contributed to a reduction in TB incidence through strategies such as directly observed treatment and the involvement of community health workers, achieving outcomes comparable to those in other developed cities. Additionally, the higher prevalence of MDR-tuberculosis in the earlier period may be partly explained by greater rates of treatment non-compliance during that time.

Regarding factors associated with resistance, prior tuberculosis treatment doubled the risk of any kind of resistance. Acquired resistances often originate from earlier, suboptimal programs or poor adherence [18], underscoring the importance of robust tuberculosis control. Globalization and migration from high-incidence regions complicate disease control, reinforcing the need for sustained efforts. Our results confirm that foreign origin raises resistance risk threefold, as reported previously in other studies [7,11,14,15,19].

Although HIV coinfection can accelerate tuberculosis progression, we did not find a significant association between HIV and resistance. Previous works on HIV and MDR-tuberculosis yield contradictory results, often influenced by nosocomial transmission in certain regions [18,20]. A Spanish study from 1995 to 2013 also found no link between HIV and anti-tuberculosis drug resistance [14].

In this study, two distinct resistance patterns were observed among second-line drugs. Ethionamide, rifabutin, and cycloserine showed the highest rates of resistance, while resistance to other second-line drugs was low or absent. It is worth noting that cycloserine is known to be chemically unstable, and in vitro resistance results may not always reflect true clinical resistance [21], but its toxicity limited its use, minimizing this problem. Rifabutin shares with rifampicin the determinant region of the *rpo*B gene where most of the mutations causing resistance are located. For this reason, at least 85% of rifampicin-resistant isolates are also resistant to rifabutin [22]. In this study, cross resistance between both was 100%.

Ethionamide deserves a special comment. In this present study, this drug was resistant in a quarter of the isolates with any kind of resistance to first-line antibiotics, mainly isoniazid, with the highest association to MDR-tuberculosis isolates, more than two times the observed in monoresistant cases (Table 4). This seems paradoxical since ethionamide and isoniazid are analogs, and both inhibit the inhA enzyme, causing inhibition of mycolic acid biosynthesis. Mutations in the promoter and structural regions of the *inh*A gene confer cross-resistance to both, being usually a low-level resistance [23,24]. However, resistance to ethionamide results from different mechanisms, including *inh*A and its promoter, the NADH-dehydrogenase encoded *ndh*, and the MShA enzyme involved in mycothiol biosynthesis, as well as mutations in the *eth*A gene that codify the activating enzyme of the prodrug and the repressor *eth*R [23,24]. This explains that the cross-resistance between both is not 100%, and, in practice, ethionamide should be tested in isoniazid-resistant isolates.

Regarding the other second-line drugs (Table 4), the resistance found was minimal. This is congruent with low-resistance incidence areas, as our country [13] is in opposition to countries with high rates of MDR-tuberculosis cases [25]. In addition, when the second-line drugs were tested against susceptible isolates to first-line antibiotics, almost no resistance was found. This is consistent with no previous exposure to these drugs.

The resistance pattern of second-line drugs underscores the necessity of continuous resistance surveillance and individualized drug susceptibility testing to optimize second-line tuberculosis treatment strategies, despite aminoglycosides, quinolones, and linezolid being the drugs of choice in addition to bedaquiline [26].

The strength of this study lies in its three decades of tuberculosis data, with almost 2500 cases including complete demographic, clinical, and microbiological data. It has public health implications, particularly in the context of Southern Europe and urban centers facing dynamic migration patterns and resistance patterns.

This study has limitations. Firstly, it includes methodological shifts in culture media and DSTs, which have improved over time and might inflate detection of resistance. However, earlier evaluations found no significant differences in phenotypic tests used [27]. Also, changes in critical drug concentrations (e.g., streptomycin, ethambutol) may partly explain some rises. Secondly, the classification of new versus previously treated cases relied on self-report, risking misclassification, especially among foreign-born patients. Finally, because this is a single-center study, generalization is limited; however, the large sample size over three decades provides sufficient statistical power and detailed microbiological, clinical, and sociodemographic data applicable to our clinical setting. We recognize that molecular determinants and molecular epidemiology would add further depth but were not included.

## 4. Materials and Methods

Epidemiological design and study population

This is a cross-sectional observational study analyzing resistance to first-line drugs and related variables between 1999 and 2022. It also describes second-line drug resistance in two groups: a subset of first-line-resistant isolates and a subset of first-line susceptible isolates.

The Hospital Clínic of Barcelona (HCB) located in Barcelona-Spain, is a tertiary care facility serving approximately 540,000 inhabitants, provided the study sample. Data from all patients with tuberculosis diagnosed at HCB from 1991 to 2022 were included.

Inclusion criteria:

1. Patients with tuberculosis confirmed microbiologically by an isolate of *Mycobacterium tuberculosis* complex and drug susceptibility test (DST) for first-line drugs. 2. Available demographic and clinical data at diagnosis. 3. Information on follow-up and treatment conclusion.

Exclusion criteria:

1. *Mycobacterium bovis* isolates. 2. *M. tuberculosis* isolates were received from other centers solely for DSTs.

Data sources:

1. Basic microbiological and clinical data were sourced from HCB’s Microbiology Department. 2. Sociodemographic and epidemiological data were provided by the Barcelona Tuberculosis Program from the Barcelona Public Health Agency. 3. The patient’s data was anonymized, and a serial number was assigned.

Variables: The main study variable was “resistance to antituberculous drugs”. Independent variables included sociodemographic, epidemiological, clinical, and microbiological characteristics as follows.

According to the number of resistant drugs and following the updated WHO guidelines [6], five resistant categories were assigned to each TB case: (1) Susceptible: susceptible to all first-line drugs; (2) Monoresistance: resistance to a single first-line drug; (3) Multiresistance: simultaneous resistance to isoniazid and rifampicin; (4) Polyresistance: resistance to two or more drugs, excluding isoniazid and rifampicin; (5) Extensively drug resistant (XDR-TB): MDR-TB with additional resistance to any fluoroquinolone and at least one Group A drug, such as bedaquiline or linezolid.

Tuberculosis cases were categorized as new (never treated or treated for fewer than 30 days) or previously treated (a 12-month gap between the end of a previous treatment and the start of a new episode) [6].

Sociodemographic variables included diagnosis period (1991–2000, high human immunodeficiency virus (HIV) seroprevalence, and 2001–2022, high immigration population), sex, age, country/region of origin, and social instability (homelessness or prison).

Clinical characteristics consider tuberculosis location (pulmonary or extrapulmonary), comorbidities (HIV infection, immunosuppression factors), and behavioral factors (smoking, alcohol use; >60 g/day in men, >40 g/day in women).

Treatment conclusions were categorized as cured, dead, moved outside Barcelona, lost during follow-up, and other (chronic disease, prolonged or failed treatment).

Mycobacterial Culture: Isolation and identification of *M. tuberculosis* from clinical samples followed standardized procedures with different media over time: 1991–2001: radiometric BACTEC 460tuberculosis^®^ (Becton-Dickinson, New Jersey, MD, USA), 2002–2022: BACTEC MGIT960^®^ (Becton-Dickinson, MD, USA).

First-line Drug Susceptibility Testing (FLDST): Performed using BACTEC 460TB^®^ (1991–2001) and BACTEC MGIT960^®^ (2002–2022), following manufacturer’s instructions. The critical drug concentrations (µg/mL) varied as follows: 1991–2001 were isoniazid 0.1 (H), rifampicin 2 (R), ethambutol 7.5 (E), streptomycin 6 (S), pyrazinamide 100 (Z). From 2002 to 2022: isoniazid 0.1, rifampicin 1, ethambutol 5, streptomycin 1 and 4, pyrazinamide 100.

Second-line Drug Susceptibility Testing (SLDST): Performed using BACTEC MGIT960^®^ (2002–2022). the critical concentrations were based on literature [28]. The drug-concentrations were (µg/mL): amikacin 1, capreomycin 2,5, cycloserine 15, ethionamide 5, isoniazid 0.4 and 1, kanamycin 1, ofloxacin 1, moxifloxacin 1, linezolid 1, and rifabutin 0.5. SLDST was performed for isolates with first-line resistance. In addition, a subset of first-line susceptible isolates was also analyzed using SLDST.

Statistical Analysis: For the purposes of the analysis, the dataset was divided into two periods (1991–2000 vs. 2001–2022). Trends in tuberculosis cases, resistance rates, and MDR-tuberculosis rates were calculated using linear regression analysis. A descriptive analysis was performed, computing frequencies and percentages for qualitative variables and medians plus interquartile ranges for quantitative ones. Percentages were compared using the chi-square test or Fisher’s exact test, and medians were compared using the Mann–Whitney U test. Bivariate and multivariate regression analysis assessed associations between drug resistance and independent variables. Odds ratios (OR), adjusted odds ratios (ORa), and their 95% confidence intervals (95% CI) were calculated. Statistical significance was set at 5%. All analyses were conducted with SPSS v21 and STATA v16.

Ethical aspects: The study followed legal regulations on data confidentiality (Organic Law 15/1999 on the Protection of Personal Data). All data were anonymized, making it impossible to identify individual participants. The study was approved by the Hospital Ethics and Research Committee (registration number/2014/0678).

## 5. Conclusions

Over this 31-year period, we observed a noticeable rise in overall tuberculosis drug resistance in the second period of the study (2000 onwards), particularly among foreign-born and previously treated patients, being three-fold and two-fold increases, respectively. These findings underscore the evolving nature of drug resistance in tuberculosis and highlight the need for continuous resistance surveillance, maintaining robust tuberculosis control programs, improving treatment adherence, and conducting continuous epidemiological surveillance.

## Figures and Tables

**Figure 1 antibiotics-14-00875-f001:**
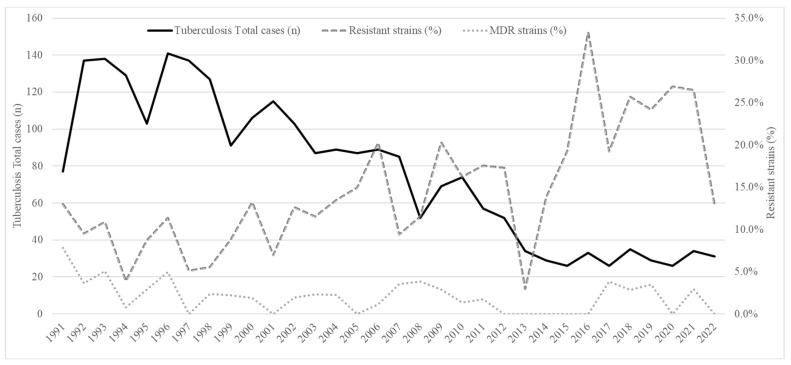
Trends of yearly tuberculosis total cases and resistance rates (overall and MDR-tuberculosis). MDR: multidrug resistance.

**Figure 2 antibiotics-14-00875-f002:**
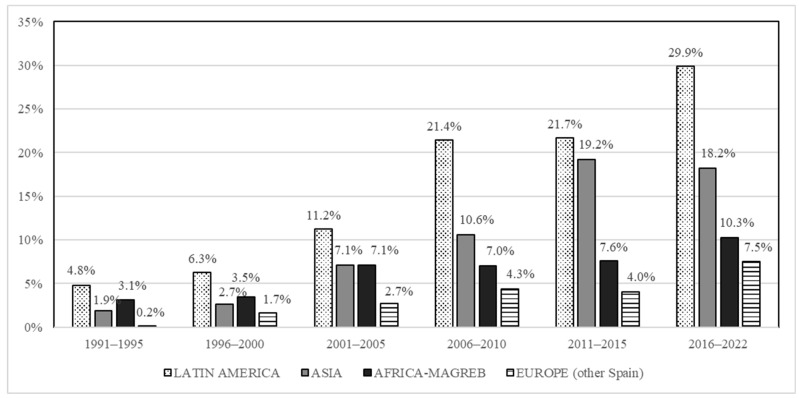
Distribution of the percentages of tuberculosis cases diagnosed in foreign-born individuals by world regions in 5-year intervals from 1991 to 2022.

**Figure 3 antibiotics-14-00875-f003:**
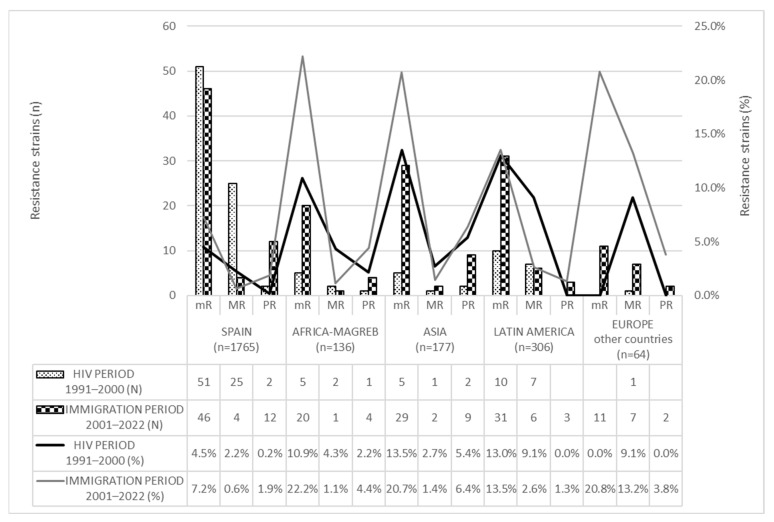
Resistance group prevalence by period and region/country. mR: monoresistance. MR: multiresistance. PR: poliresistance.

**Table 1 antibiotics-14-00875-t001:** Demographic, clinical, and microbiological data of tuberculosis cases.

	Total (N = 2448)		1991–2000 (N = 1301)		2001–2022 (N = 1147)		*p*-Value
Gender (male), N (%)	1715	70.1	956	73.4	759	66.1	<0.001 ^1^
Age (median [IQR]), years	38	29–55	36	29–53	40	29–58	<0.001 ^2^
Age distribution, N (%)							
<18	42	1.7	20	1.5	22	1.9	0.007 ^1^
18–29	610	24.9	339	26.1	271	23.6	
30–49	1055	43.1	586	45	469	40.9	
>49	741	30.3	356	27.4	385	33.6	
Origin, N (%)							
Spain	1765	72.1	1130	86.9	635	55.4	<0.001 ^1^
Rest of Europe	64	2.6	11	0.8	53	4.6	
Africa–Maghreb	136	5.6	46	3.5	90	7.8	
Latin America	306	12.5	77	5.9	229	20	
Asia	177	7.2	37	2.8	140	11.2	
Social situation, N (%)							
Homeless people	190	7.8	87	6.7	103	9	<0.021 ^1^
Prison	257	11.4	223	17.1	34	3	<0.001 ^1^
HIV infection, N (%)	605	24.7	450	34.6	155	13.5	<0.001 ^1^
Other clinical disorders, N (%)							
Kidney failure	69	2.8	34	2.6	35	3.1	0.29 ^1^
Immunosuppression	108	4.4	29	2.2	79	6.9	<0.001 ^1^
Diabetes	148	6.0	66	5.1	82	7.1	0.020 ^1^
Alcohol consuming	426	17.4	259	19.9	167	14.6	<0.001 ^1^
Tuberculosis clinical location, N (%)							0.883 ^1^
Pulmonary	1960	80.1	1045	80.3	915	79.8	0.38 ^1^
Extra-pulmonary	488	19.9	256	19.7	232	20.2	0.38 ^1^
Bacteriology, N (%)							<0.001
Culture +/smear −	1194	48.8	556	42.7	638	55.6	
Culture +/smear +	1254	51.2	745	57.3	509	44.4	
New tuberculosis cases, N (%)	2189	89.4	1110	85.3	1079	94.1	<0.001 ^1^
Non-susceptible tuberculosis, N (%)	299	12.2	112	8.6	187	16.3	<0.001 ^1^
Outcome, N (%)							<0.001 ^1^
Treatment completed	1537	62.8	740	56.9	797	69.5	
Dead	279	11.4	182	14	97	8.5	
Off Barcelona ^3^	425	17.4	311	23.9	114	9.9	
Lost to follow-up	196	8	65	5	121	11.4	
Another ^4^	11	0.4	3	0.2	8	0.7	

IQR: interquartile range; +: Positive, −: Negative; ^1^ X^2^ test to compare percentages between periods; ^2^ U Mann–Whitney test to compare medians between periods; ^3^ Patients moved out from Barcelona; ^4^ Chronic cases or long treatment.

**Table 2 antibiotics-14-00875-t002:** Tuberculosis cases. Distribution according to drug resistance and period.

	Total (N = 2448)		1991–2000 (N = 1301)		2001–2022 (N = 1147)		*p*-Value
	N	%	N	%	N	%	
New tuberculosis cases	2189	89.4	1110	85.3	1079	94.1	
Treated tuberculosis cases	259	10.6	191	14.7	68	5.9	<0.001
Total cases							
Susceptible	2149	87.8	1189	91.4	960	83.7	
Resistant	299	12.2	112	8.6	187	16.3	<0.001
New cases							
Susceptible	1944	88.8	1035	93.2	909	84.2	
Resistant	245	11.2	75	6.8	170	15.8	<0.001
Treated cases							
Susceptible	205	79.2	154	80.6	51	75	
Resistant	54	20.8	37	19.4	17	25	0.42
Resistance ^1^	414	16.9	168	12.9	246	21.4	<0.001
H	190	7.8	94	7.2	96	8.4	
R	59	2.4	36	2.8	23	2	
E	24	1	7	0.5	17	1.5	
Z	58	2.4	22	1.7	36	3.1	
S	83	3.4	9	0.7	74	6.5	
Monoresistance	208	8.5	71	5.5	137	11.9	<0.001
H	102	4.2	53	4.1	49	4.3	
R	3	0.1	0	-	3	0.1	
E	8	0.3	0	-	8	0.7	
Z	29	1.2	8	0.6	21	1.8	
S	66	2.7	10	0.8	56	4.9	
Multiresistance	56	2.3	36	2.8	20	1.7	0.498
H + R	23	0.9	16	1.2	7	0.6	
H + R + E	2	0.1	2	0.2	0	-	
H + R + Z	9	0.4	8	0.6	1	0.1	
H + R + S	6	0.2	4	0.3	2	0.2	
H + R + E + Z	3	0.1	2	0.2	1	0.1	
H + R + E + S	3	0.1	1	0.1	2	0.2	
H + R + Z + S	5	0.2	2	0.2	3	0.3	
H + R + E + Z + S	5	0.2	0	0.0	5	0.4	
Polyresistance	35	1.4	5	0.4	30	2.6	0.201
H + E	1	0.1	1	0.1	0	0.0	
H + Z	3	0.1	1	0.1	2	0.2	
H + S	25	1.0	3	0.2	22	1.9	
H + E + S	2	0.1	0	-	2	0.2	
H + Z + S	1	0.1	0	-	1	0.1	
S + Z	3	0.1	0	-	3	0.1	

E: Ethambutol; H: Isoniazid; Z: Pyrazynamide R: Rifampicin; S: Streptomycin. ^1^ Including monoresistant, multiresistant, and polyresistant isolates.

**Table 3 antibiotics-14-00875-t003:** Statistical analysis of variables associated with resistance and multiresistance.

	Any Kind of Resistance ^1^		Multiresistance ^1^	
	Bivariate	Multivariate	Bivariate	Multivariate
	OR (95% CI)	aOR (95% CI)	OR (95% CI)	aOR (95% CI)
Period				
1990–2000	1	−	1	1
2001–2022	1.63 (1.28–2.09)	NS	0.53 (0.30–0.94)	0.40 (0.23–0.69)
Gender				
Man	1	−	1	−
Woman	1.15 (0.89–1.47)	NS	1.34 (0.99–1.81)	NS
Age (years)				
	1.00 (0.99–1.01)	NS	1.00 (0.99–1.01)	NS
Origin				
Spain	1	1	1	1
Rest of Europe	3.45 (2.48–4.80)	3.46 (2.42–4.95)	3.00 (1.89–4.75)	3.14 (1.94–5.07)
Africa–Maghreb	3.33 (2.18–5.09)	3.45 (2.28–5.21)	3.70 (2.22–6.17)	3.76 (2.25–6.29)
Latin America	2.59 (1.90–3.51)	2.66 (1.92–4.59)	2.72 (1.87–3.96)	2.69 (1.82–3.97)
Asia	2.81 (1.82–4.33)	2.97 (2.51–7.92)	2.91 (1.69–4.98)	2.89 (1.68–4.98)
Tuberculosis case				
New	1	1	1	1
Previously treated	2.09 (1.59–2.75)	2.56 (1.96–3.35)	1.41 (0.94–2.13)	1.79 (1.19–2.68)
Location				
Pulmonary	1	−	1	−
Extra-pulmonary	0.94 (0.70–1.27)	NS	1.16 (0.82–1.64)	NS
Bacteriology				
Culture +/smear −	1	−	1	−
Culture +/smear +	1.05 (0.83–1.33)	NS	0.91 (0.68–1.22)	NS
HIV infection				
Negative	1	−		−
Positive	0.85 (0.64–1.13)	NS	0.77 (0.54–1.10)	NS
Unknown	0.74 (0.51–1.06)	NS	0.65 (0.41–1.04)	NS
Homeless people				
Yes	0.79 (0.46–1.34)	NS	0.80 (0.42–1.54)	NS
No	1	-	1	-
Prison				
Yes	0.55 (0.34–0.90)	0.57 (0.35–0.94)	0.30 (0.13–0.67)	0.33 (0.15–0.74)
No	1	1	1	1
Alcohol consuming				
Yes	0.72 (0.51–1.02)	NS	0.73 (0.48–1.12)	NS
No	1	-	1	-
Immunosuppression				
Yes	0.77 (0.60–0.99)	NS	0.77 (0.56–1.04)	NS
No	1	-	1	-

OR: Odds ratio; aOP: adjusted Odds ratio; CI: confidence interval; +: Positive, −: Negative; ^1^ compared to drug-susceptible cases.

**Table 4 antibiotics-14-00875-t004:** Second-line antituberculous drug resistance and distribution according to the category of first-line drug resistance.

			Distribution According First-Line Drug Resistance					
Second-Line Drug			Monoresistant		Polyresistant		Multiresistant	
	Tested ^1^ N	Resistant ^1^ (%)	R/C ^2^	% ^2^	R/C ^2^	% ^2^	R/C ^2^	% ^2^
Rifabutin	99	14 (14.1)	1/66	1.5	0/20	0	13/13	100
Ethionamide	200	49 (24.5)	21/124	16.9	10/33	30.3	18/43	41.9
Ofloxacin	202	8 (4)	5/125	4	0/33	0	3/44	6.8
Amikacin	202	2 (1)	0/125	0	0/33	0	2/44	4.5
Capreomycin	198	2 (1)	1/123	0.8	0/33	0	1/42	2.4
Cycloserine	201	26 (12.9)	18/125	14.4	0/33	0	8/43	18.6
Kanamycin	203	2 (1)	0/125	0	0/33	0	2/45	4.4
Linezolid	57	0 (0)	0/41	0	0/6	0	0/10	0

^1^ For each second-line drug; times tested, number, and % of resistance. ^2^ R/C; second-line drug resistant/cases with first-line drug resistance of each category (monoresistant, polyresistant, and multiresistant).

## Data Availability

Not applicable.

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
