# Peer review of "Trends in Antituberculosis Drug Resistance and Associated Factors: A 31-Year Observational Study at a Tertiary Hospital in Barcelona"

_antibiotics, 2025, doi:10.3390/antibiotics14090875_

Round 1

Reviewer 1 Report

Comments and Suggestions for Authors

The authors of the peer-reviewed manuscript analysed the evolution of drug resistance to anti-tuberculosis drugs over 31 years of observation. They demonstrated a noticeable increase in overall resistance, particularly among foreign-born and previously treated patients. The data presented in the manuscript underscore the evolving nature of drug resistance in tuberculosis and the global need for continuous resistance monitoring. Furthermore, they emphasise the need to improve tuberculosis control programs, adherence to treatment recommendations, and ongoing epidemiological surveillance. The manuscript is written according to journal guidelines, with the introduction presenting key information regarding the epidemiology and treatment of tuberculosis. The methodology is well-written, avoiding unnecessary information. The results section is clear and includes the most important data, and the discussion section compares the findings with data from other centres. The authors also describe the limitations of this study. The conclusions are well-formulated, and the references are relevant to the content.

Author Response

We thank the reviewer for the positive evaluation and valuable comments. No further modifications were required.

Reviewer 2 Report

Comments and Suggestions for Authors

The research conducted by You All is good.

What is the main question addressed by the research? The main question addressed by the research is which is the evolution of resistance to the first and second-line antituberculosis drugs in Barcelona over 31 years (1991-2022), focusing on resistance patterns and sociodemographic factors influencing its evolution. • Do you consider the topic original or relevant to the field (al campo)?. Yes, is original and relevant to the field.  Does it address a specific gap in the field?. No. Please also explain why this is/ is not the case.  scientific research is well done: The scientific research is well done. • What does it add to the subject area compared with other published
material?. It add scientific studies of the tuberculosis cases in Barcelona, and the scientific studies indicates that tuberculosis cases has decreased over the past three decades, and antibiotic resistance has increased in Barcelona by foreign-born and retreatment cases, and the effect of treatments on patients with tuberculosis in Barcelona. • What specific improvements should the authors consider regarding the
methodology? . The effect of medical treatments, and the reason for human resistance to tuberculosis. • Are the conclusions consistent with the evidence and arguments presented
and do they address the main question posed?. Yes, because of the medical investigations and analysis performed. • Are the references appropriate?. Yes • Any additional comments on the tables and figures?. Yes The medical and scientific analysis performed regarding the cases of Tuberculosis in Barcelona has been performed Good.

Author Response

(The authors gave the same response as above.)

Reviewer 3 Report

Comments and Suggestions for Authors

Comments for antibiotics-3823522-peer-review-v1

The authors have conducted interesting research on antitubercular drug resistance observed from a Barcelona based hospital

This manuscript can be accepted after minor corrections.

  1. Side effects of used antitubercular drugs in studies in no resistant patients and resistance patients can be discussed if clinically investigated for a 6/9/12/24 months drug treated patients.
  2. Patients’ behaviours (short term/long term) can be discussed if clinically investigated for a 6/9/12/24 months drug treated patients.
  3. Conclusion is not sufficient. It can include what can be treatment approaches to minimise the antitubercular drug resistance, based on a 31-year study. Comparison among antitubercular drugs which were found to be best in this current study. Which antitubercular drugs could be preferable in a case of mono/multi/poly resistance.
  4. Any toxicity related to any antitubercular drug after a long-term treatment can be mentioned.
  5. An English language edit/correction is essential before publications.

Comments on the Quality of English Language

An English language edit/correction is essential before publications.

Author Response

COMMENT 1. Side effects of used antitubercular drugs in studies in no resistant patients and resistance patients can be discussed if clinically investigated for a 6/9/12/24 months drug treated patients.

RESPONSE 1. We agree that analysing treatment-related side effects would provide valuable clinical insights. Unfortunately, such data were not collected in our database, and therefore we cannot address this aspect in the present study, which is focused on describing the distribution of tuberculosis resistance along the period within the different populations and its risk factors.

COMMENT 2. Patients’ behaviours (short term/long term) can be discussed if clinically investigated for a 6/9/12/24 months drug treated patients.

RESPONSE 2. We believe the reviewer is referring to treatment outcomes. This information is presented in Table 1, and we have now added a clarifying paragraph in the Results section to highlight improvements in treatment completion, mortality, and follow-up over time. Lines 111-115; “Between 1991–2000, 56.9% of patients completed treatment, compared with 69.9% in 2001–2022 (p < 0.001). Mortality decreased from 17.4% to 9.7% (p < 0.001), and both treatment abandonment and loss to follow-up were also significantly reduced in the later period. Overall, these results demonstrate a marked improvement in treatment outcomes over time.

COMMENT 3. Conclusion is not sufficient. It can include what can be treatment approaches to minimise the antitubercular drug resistance, based on a 31-year study. Comparison among antitubercular drugs which were found to be best in this current study. Which antitubercular drugs could be preferable in a case of mono/multi/poly resistance.

RESPONSE 3. We appreciate this thoughtful suggestion. However, our dataset did not include detailed information on individual treatment regimens, so we cannot draw reliable conclusions on optimal drug choices in mono, multi, or polyresistance. However, most patients would have probably followed the standard treatments following the WHO recommendations. We have therefore kept the conclusions focused on resistance trends and risk factors.

COMMENT 4. Any toxicity related to any antitubercular drug after a long-term treatment can be mentioned.

RESPONSE 4. As exposed before, we agree that analysing treatment-related toxicity would provide valuable clinical insights. Unfortunately, such data were not collected in our database, and therefore we cannot address this aspect in the present study.

COMMENT 5. An English language edit/correction is essential before publications.

RESPONSE 5. In response to this recommendation, the manuscript has been revised by a native English speaker with experience in scientific publications. Minor language edits were implemented.

Reviewer 4 Report

Comments and Suggestions for Authors

Dear Authors, 

the presented study can provide some interesting insights, but a few improvements should be taken into consideration: 

there are some unexplained abbreviations in the Abstract, which should be explained when first mentioned in the text.

Introduction

  1. 57-59: this sentence is unclear – it can mean both that the mentioned countries achieved or did not achieve the 50% reduction. Please rephrase
  2. 80-82: this sentence/fragment sounds disconnected from the rest of the Introduction. Either incorporate it smoothly into the text or remove it. How does it connect to the context of the study? If the Authors want to mention the identification of drug resistance, they need to elaborate on it and refer it to their study.
  3. 83-85: even though the strict aim of this study has been provided, this fragment lacks information about the actual scientific or practical outcome of this study. Please expand this fragment.

Results

Table 1: I suggest that the remarks determined with upper-case numbers (e.g. Off Barcelona, 4) should be explained directly by the table.

some figures (e.g. Fig. 2, 3) should have error bars or confidence intervals.

Ensure consistent use of terms like “monoresistance,” “polyresistance,” and “multiresistance.” Definitions are provided but could be reiterated briefly in the discussion for clarity.

The manuscript mentions that the increase in resistance is statistically significant, but the multivariate model for “any resistance” in the second period is marked as “NS” (not significant). Clarify this apparent discrepancy.

Methods: The criteria for defining drug resistance (MDR, XDR, mono-resistance) should be clearly aligned with WHO definitions, specifying any changes over the 31-year period.

general remark – the term „any resistance” occurs quite frequently throughout the text, but it doesn’t seem correct or understandable in some places. Consider replacing it.

Conclusions – I appreciate not repeating the results in the Conclusions section, but a brief summary of the outcomes would be beneficial.

Comments on the Quality of English Language

The overall quality of English language is fine and the text reads generall well. There are a few issues e.g. with terminology like "any resistance" that should be replaced, but nothing major. 

Author Response

COMMENT 1. There are some unexplained abbreviations in the Abstract, which should be explained when first mentioned in the text.

RESPONSE 1. We appreciate the reviewer’s remark. Since OR and CI are widely used statistical terms, we kept them without definition in the Abstract, but both are clearly defined in the Abbreviations and Methods sections.

COMMENT 2. Introduction, lines 57-59: this sentence is unclear – it can mean both that the mentioned countries achieved or did not achieve the 50% reduction. Please rephrase

RESPONSE 2. We have rephrased the sentence for clarity. Lines 57–59: “Despite this, achieving the 50% reduction goal by 2025 in Spain remains unlikely, as it happens in other Western European countries, including Portugal, Italy, France, and Belgium

COMMENT 3. Introduction, lines 80-82: this sentence/fragment sounds disconnected from the rest of the Introduction. Either incorporate it smoothly into the text or remove it. How does it connect to the context of the study? If the Authors want to mention the identification of drug resistance, they need to elaborate on it and refer it to their study.

RESPONSE 3. We agree with the reviewer that the original phrasing was not well connected. We have revised the paragraph and moved it earlier in the Introduction to better highlight its relevance to our study. Lines 68-71: “An important aspect to reach this goal is the surveillance of drug resistance M.tuberculosis strains that is based on phenotypic drug susceptibility testing as the reference standard. This will allow to know the epidemiological situation of TB resistance and get the best treatment for patients.

COMMENT 4. Introduction, lines 83-85: even though the strict aim of this study has been provided, this fragment lacks information about the actual scientific or practical outcome of this study. Please expand this fragment.

RESPONSE 4. As suggested, we have expanded the aim to include the practical implications of our findings. Lines 85-87: “The results would help to know the TB control program outcomes and identify TB resistance risk factors and improve its management.

COMMENT 5. Results, Table 1: I suggest that the remarks determined with upper-case numbers (e.g. Off Barcelona, 4) should be explained directly by the table.

RESPONSE 5. As suggested, all the remarks have been explained at the bottom of Table 1. Line 108-109: “1 X2 test to compare percentages between periods. 2 U Mann-Whitney test to compare median between periods. 3 Patients moved out from Barcelona.. 4 Chronic cases or long treatment.”

COMMENT 6. Results, some figures (e.g. Fig. 2, 3) should have error bars or confidence intervals.

RESPONSE 6. We thank the reviewer for this thoughtful suggestion. Figures 2 and 3 were designed as descriptive illustrations of aggregated case counts and resistance proportions across the study period. Confidence intervals and statistical comparisons are already provided in the Results section and in Table 3. Since some subgroups have very small sample sizes, adding error bars or confidence intervals to the figures would not provide additional clarity and could be misleading for interpretation. For these reasons, we have kept Figures 2 and 3 as descriptive. We hope the reviewer agrees that this presentation remains clear and appropriate.

COMMENT 7. Results, Ensure consistent use of terms like “monoresistance,” “polyresistance,” and “multiresistance.” Definitions are provided but could be reiterated briefly in the discussion for clarity.

RESPONSE 7. We reviewed the terminology and ensured consistent use throughout. For clarity, we added a brief explanation in the Discussion. Lines 222-224:” Among resistance categories, monoresistance (one single first-line drug) was most frequent, followed by MDR-tuberculosis (H and R drug resistance) and polyresistance (two or more drugs resistance other than MDR).

COMMENT 8. Results, The manuscript mentions that the increase in resistance is statistically significant, but the multivariate model for “any resistance” in the second period is marked as “NS” (not significant). Clarify this apparent discrepancy.

RESPONSE 8. We clarified in the Results section that the increase in resistance after 2000 was statistically significant in the bivariate analysis (OR 1.63) but lost significance in the multivariate model, likely due to confounding with foreign-born status and previous treatment history. Lines 166-168: “Despite an increase of TB resistance cases after 2000, with a 1.63-fold higher risk, it was not statistical significance in the multivariate analysis probably influenced by the foreign-born status and previous TB treatment. (Table 3).

COMMENT 9. Methods, The criteria for defining drug resistance (MDR, XDR, mono-resistance) should be clearly aligned with WHO definitions, specifying any changes over the 31-year period.

RESPONSE 9. We have classified TB cases within the resistant categories following the last updated WHO classification document (reference nº6). The XDR classification was not defined in the manuscript due that we haven’t had any case. However, we have added in the “Materials and Methods” section the XDR category. Lines 318-324: “According to the number of resistant drugs and following the updated WHO guidelines (6), five resistant categories were assigned to each TB case. 1) Susceptible: susceptible to all first-line drugs; 2) Monoresistance: resistance to a single first-line drug; 3) Multiresistance: simultaneous resistance to isoniazid and rifampicin; 4) Polyresistance: resistance to two or more drugs, excluding isoniazid and rifampicin; 5) Extensively drug resistant (XDR-TB): MDR-TB with additional resistance to any fluoroquinolone and at least one Group A drug, such as bedaquiline or linezolid

COMMENT 10. General remark – the term „any resistance” occurs quite frequently throughout the text, but it doesn’t seem correct or understandable in some places. Consider replacing it.

RESPONSE 10. The term “any resistance” is used along the text describing any kind of TB drug resistance without differentiating from the terms of monoresistance, polyresistance and multiresistance. Following the reviewer suggestion we have added “any kind of resistance” throughout the text for greater clarity. Lines: 238,258, Table 3 line 170

COMMENT 11. Conclusions – I appreciate not repeating the results in the Conclusions section, but a brief summary of the outcomes would be beneficial.

RESPONSE 11. We have added the following sentence to highlight the main results in the “Conclusions” section. Line 367-369: “Over this 31-year period, we observed a noticeable rise in overall tuberculosis drug resistance in the second period of the study (2000 onwards), particularly among foreign-born and previously treated patients, being three-fold and two-fold increases respectively.

COMMENT 12. The overall quality of English language is fine and the text reads generall well. There are a few issues e.g. with terminology like "any resistance" that should be replaced, but nothing major.

RESPONSE 12. We have sent the manuscript to an English native reviewer with experience in scientific publications and minor language modifications have been done. The term “any resistance” has been clarified in point 10.